# Distributional effects of parental time investments on children's socioemotional skills and nutritional health

**Juan Carlos Caro** *

Department of Industrial Engineering, Universidad de Concepcion, Concepcion, Chile

* juancaros@udec.cl

**Data Availability Statement:** Data can be access by direct request to JUNAEB: https://www.junaeb.cl/.

**Funding:** This work was partially funded by National Agency for Research and Development of

## Abstract

Parental behavior is paramount to child health and skill formation, explaining a significant portion of differences in developmental outcomes. However, little is known regarding the distributional effects of parental time allocation at different levels of children's outcomes. I use a national administrative dataset of Chilean pre-school students to the estimate production functions for socioemotional development and body mass index z-scores at every decile of the distribution at baseline. Modest average effects conceal significant heterogeneity on the returns to parental time investments. Children in the bottom of the socioemotional development distribution could gain up to 0.4 standard deviations for a one standard deviation increase in time investments. A similar increase can lead to a reduction of 0.8 standard deviations in body mass index among severely obese students. Evidence reveals that children with high developmental scores are unlikely to benefit from additional parenting time.

## Introduction

Inadequate nutrition and psychosocial stimulation prevent nearly one of every two children from reaching their lifetime human capital potential worldwide [1–3]. Evidence shows that parental time investments play are paramount to reduce gaps in child development [4–7]. Parental time investments, also defined as educational child care [8], refer to any activities that promote child development, beyond basic care (e.g. hygiene, food, shelter) and supervision. The latter could include stimulation, socialization and physical activities. Overall, studies from high- and middle-income countries support the skill formation model introduced by [9], estimating significant returns to parental time investments, particularly in the first years of life. To date, research have focused on the average partial effects of parental time towards skill accumulation, thus any distributional effects are determined by the functional form imposed in the model. The latter assumes an unique skill formation technology, and differences in returns to parental investments based on child skills are solely determined by the interaction between inputs and investments. However, recent evidence on the distributional effects of nutritional and educational investments reveals important heterogeneity on the marginal returns at different quantiles of the skill distribution [10–12]. For example, [12] shows that the returns on health and cognition investments vary significantly depending on the decile of child skills at

Chile (ANID) through grant PAI/INDUSTRIA 79090016. The funders had no role in study design, data collection and analysis, decision to publish, or preparation of the manuscript.

**Competing interests:** The authors have declared that no competing interests exist.

baseline. The latter suggests that the marginal productivity of parental time investments could also vary significantly based on the child development at a given period.

The degree of heterogeneity on the returns to parental time investments at different levels of child nutritional health and socioemotional skills remains as an empirical issue. As noted earlier, previous studies have focused on the potential substitutability or complementarity between parental time and prior skills, assuming the parameters of the production function are the same for all children [4, 13, 14]. Moreover, estimation approaches often assume functional forms which impose some complementarity between inputs and parental resources. As noted by [9], the technology of skills production could be significantly nonlinear, therefore relying on the average partial effects alone is insufficient to fully characterize the heterogeneity of parental investment returns across the skill distribution. For example, is possible that time investments and prior skills are complements on average (either empirically or by construction), but investments are no longer effective (or harmful) when the model is estimated at extreme quantiles.

While limited, evidence on the distributional effects of parental time investments on prior child skills is useful to shed light on the potential nonlinearities on the skill production technology. When the interaction between skills and investments is estimated directly, results indicate that parental time is most effective for children with low levels of socioemotional skills [5, 15]. For nutritional health, results are more nuanced. While there is no study reporting directly on the effectiveness of time investments, evidence suggests compensation behavior for health deficits among families with higher socioeconomic status and educational background [16, 17]. Observed differences in parental time investments between households could be explained by heterogeneity on parents' knowledge and beliefs regarding the returns to investments [18, 19]. Altogether, is reasonable to expect that for young children, additional parenting time is more effective for those with deficits on nutritional health and socioemotional skills. Given the diminishing returns to repetition in skill formation, an additional hour on stimulation or physical activities is expected to have a smaller impact on children with already high skill levels. This argument applies does not necessarily applies to older children (facing more complex socioemotional challenges) or to other skill domains (e.g. cognition). Many dimensions of child development are progressive, thus children with high skill levels are expected to benefit more from additional parenting time for a given dimension. Similarly, the underlying processes that characterize children's nutritional health and socioemotional skills are markedly distinct before and after puberty [20, 21].

This study contributes to the current literature on parental behavior and child outcomes with novel evidence on the distributional effects of parental time investments on socioemotional development and nutritional status in a context of high-income country with increasing obesity prevalence. The model is estimated using rich administrative data from the Chilean National Board of School Aid and Scholarships (JUNAEB, Spanish acronym). The case of Chile is relevant for two key factors. First, it captures the current status of many middle- and high-income countries facing dramatically high rates of obesity prevalence in recent decades. Secondly, JUNAEB administrative data is an unique dataset, which allows to capture measures on child development, parental time investments, and nutritional status for students from all schools receiving any public funding, nearly 90% of total enrollment in the country.

The analysis follows a cohort of children that started Pre-Kindergarten in 2015, with repeated measurements at Kindergarten and First grade (nearly 200,000 students across 10,000 schools). First, I estimate a measurement system to obtain latent factors for parental time investments, parenting styles and socioemotional development, in order to address measurement error over multiple indicators [22]. Second, I use the predicted factors, as well as child and household characteristics to estimate the determinants of parental time allocation.

As in similar studies, I use local prices as instruments for parental time investments, particularly cost and quality of the relevant pre-school market [12]. Next, based on the approach proposed by [23], I estimate the production functions of socioemotional development (SED) and body mass index z-scores (BAZ) at each decile using the control function approach in both stages. This strategy allows measurement of the effects of time investments in human capital accumulation through the SED and BAZ distributions.

There are three main results. First, the estimated measurement system provides a single latent socioemotional factor, that could be interpreted as emotional stability, defined as consistency and predictability in emotional reactions in the Big Five Model [24]. Emotional stability is associated with externalizing behavior, locus of control, and self-efficacy. When comparing students based on their schools' vulnerability, the inequality in human capital accumulation increases between grades for socioemotional development but not for BAZ. Second, social support and parental self-efficacy are key determinants of time investments across households (once accounting for endogeneity). Moreover, results suggests potential complementarities between time and material investments. The results also show no differences in time allocation by labor force status of the mother, consistent with previous studies [25]. Third, the impact of parental time investments on socioemotional development and BAZ is modest, on average. However, for children with limited socioemotional development and high BAZ (those obese and severely obese), increasing time investments by one standard deviation (SD) can lead to an increase of socioemotional development of 0.4 SD and a reduction of BAZ of 0.8 SD. However, for children at the top of the socioemotional development distribution, additional time investments can lead to lower socioemotional development in the next period.

## Materials and methods

### The JUNAEB administrative data

JUNAEB is a public agency responsible for assessing students' needs and allocating resources through different programs. The main dataset follows a cohort of all Chilean children who attended Pre-K in 2015 until first grade of primary school, excluding those who attend private schools (less than ten percent of total enrollment). Teachers measure and collect information on children's height and weight, while parents provide a comprehensive household background regarding family composition, children's health and behavior, financial resources and parenting practices. S1 Appendix details the information contained in the JUNAEB data. The analytical sample includes only children measured every grade, roughly two thirds all students at baseline. Missing data, in order of importance, is due to absences at the day of measurement in one or more grades, repeating first grade, skipping one year between Kindergarten and first grade, and children not attending Pre-K or Kindergarten. I exclude students that report chronic illness or disabilities and those with implausible weight and height measurements. The total number of excluded observations represents less than 2% of the raw data.

Table 1 shows descriptive statistics of the analytical sample compared with complete cohort data in each grade. There are not significant differences in the anthropometric or household data between the children with complete information every grade versus those that missed school during measurement in at least one grade. Estimates in this study are conducted over complete case analysis. S3 Appendix contains a sensitivity analysis using Inverse Propensity Weighting (IPW) from a Probit model to predict the probability of attrition between grades. Observable variables weakly predict attrition and IPW weighted estimates are fairly similar to those of unweighted estimates. Nearly half of children are overweight, and their individual and household characteristics are rather stable over time. One exception is labor force participation among mothers, which increases ten percent points once children enter Kindergarten. Relative

**Table 1. Descriptive statistics.**

| | Pre-Kinder | | Kindergarten | | First grade | |
|---|---|---|---|---|---|---|
| **Anthropometrics and behavior** | **All** | **Sample** | **All** | **Sample** | **All** | **Sample** |
| Age (months) | 56.2 | 56.3 | 67.5 | 67.4 | 77.4 | 77.7 |
| | *4.4* | *4.4* | *4.5* | *4.3* | *4.9* | *3.8* |
| Height-for-age (Z-score) | 0.15 | 0.15 | 0.16 | 0.17 | 0.22 | 0.24 |
| | *1.2* | *1.2* | *1.19* | *1.16* | *1.1* | *1.06* |
| BMI-for-age (Z-score) | 0.97 | 0.96 | 1.03 | 1.02 | 1.00 | 1.00 |
| | *1.46* | *1.45* | *1.42* | *1.4* | *1.37* | *1.34* |
| Fraction overweight | 49.0% | 48.6% | 52.0% | 50.5% | 50.0% | 49.0% |
| Hard to understand others (%) | 16.9% | 16.1% | 16.9% | 16.0% | 18.8% | 17.0% |
| Hard to control behavior (%) | 40.0% | 39.5% | 38.5% | 37.6% | 38.7% | 38.9% |
| Hard to get along with peers (%) | 21.2% | 20.8% | 20.4% | 19.5% | 21.5% | 20.1% |
| **School characteristics** | | | | | | |
| School vulnerability index (IVE) | 69.3 | 69.4 | 69.0 | 69.4 | 69.2 | 69.4 |
| | *17.4* | *17.4* | *17.2* | *17.2* | *16.9* | *16.9* |
| Public school = 1 | 0.67 | 0.66 | 0.64 | 0.64 | 0.43 | 0.41 |
| Attended daycare = 1 | 0.71 | 0.70 | 0.72 | 0.70 | 0.71 | 0.70 |
| **Household characteristics** | | | | | | |
| Mother's schooling (years) | 12.9 | 12.6 | 12.9 | 12.8 | 12.9 | 12.7 |
| | *3.0* | *3.4* | *3.0* | *3.5* | *3.1* | *3.5* |
| Father's schooling (years) | 12.8 | 12.4 | 12.9 | 12.5 | 12.8 | 12.4 |
| | *3.1* | *3.8* | *3.1* | *3.8* | *3.2* | *3.9* |
| Mother's age (years) | 31.4 | 31.4 | 32.3 | 32.3 | 33.1 | 33.1 |
| | *6.8* | *6.8* | *6.8* | *6.8* | *6.8* | *6.8* |
| Household size | 4.6 | 4.6 | 4.7 | 4.7 | 4.7 | 4.7 |
| | *1.7* | *1.7* | *1.7* | *1.7* | *1.7* | *1.7* |
| Mother in labor force = 1 | 0.54 | 0.54 | 0.65 | 0.67 | 0.64 | 0.68 |
| Lives with father = 1 | 0.68 | 0.68 | 0.65 | 0.66 | 0.62 | 0.63 |
| Lives in urban area = 1 | 0.09 | 0.09 | 0.10 | 0.09 | 0.11 | 0.11 |
| Ethic ancestry = 1 | 0.12 | 0.11 | 0.13 | 0.12 | 0.14 | 0.13 |
| Sample size | 153,516 | 126,738 | 190,752 | 126,738 | 219,518 | 126,738 |

Table notes JUNAEB indicates anthropometric data and household survey data from the Nutritional Map. IVE indicates the Spanish acronym for the School Vulnerability Index. The panel indicates children in urban households matched with Kindergarten and Pre-Kinder data. The fraction with behavioral difficulties represents all those parents that reported any hardship (from mild to extreme). Daycare refers to children 2–4 years old. Standard deviations in italics, if applicable.

to behavioral difficulties (proxies for socioemotional development), over half of all children report at least some type of hardship, particularly to control behavior.

In order to compare across households by socioeconomic status, I constructed deciles based on the school vulnerability index calculated by JUNAEB (IVE, Spanish acronym). The IVE measures the fraction of households experiencing multidimensional poverty in a given school. Fig 1 shows the obesity prevalence and fraction of children with behavioral difficulties by grade, sex and decile of IVE. Overall, health outcomes are consistently worse for boys than girls, and differences are stable over time. Obesity is a widespread phenomenon, only girls attending less vulnerable school exhibit significantly lower average obesity rates. By first grade of school, one of every four boys and one of every five girls is obese. Similarly, children attending to schools with low vulnerability exhibit significantly lower risk of behavioral control symptoms. Fig 2 shows the distribution of parental time investments for each activity included

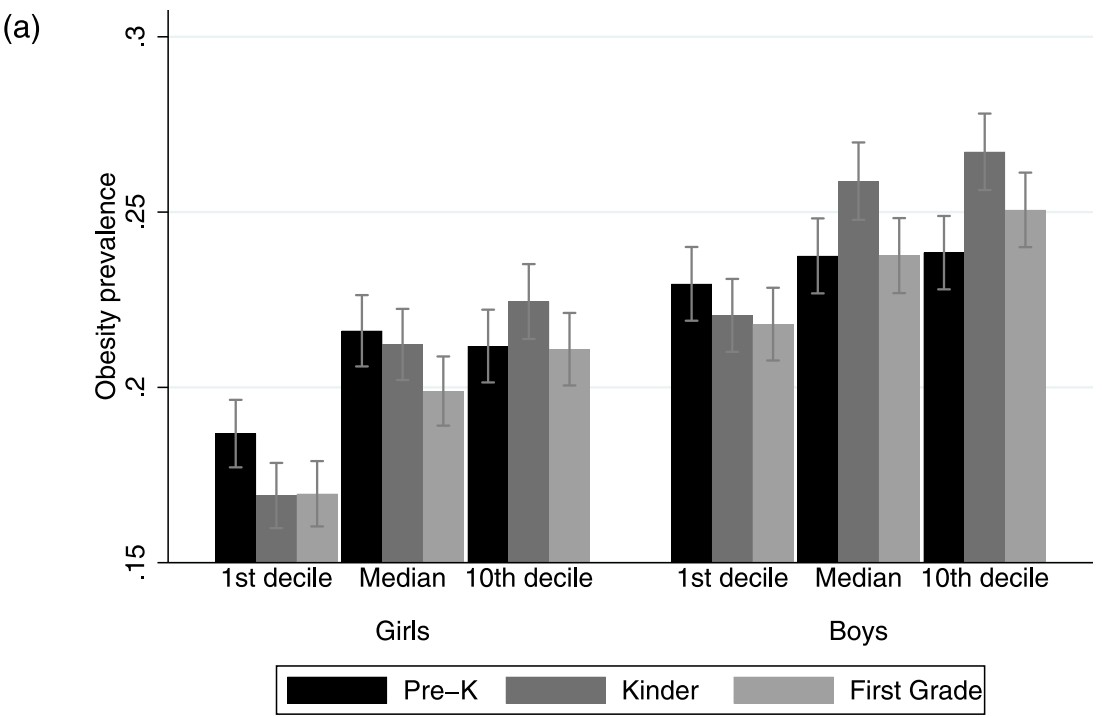

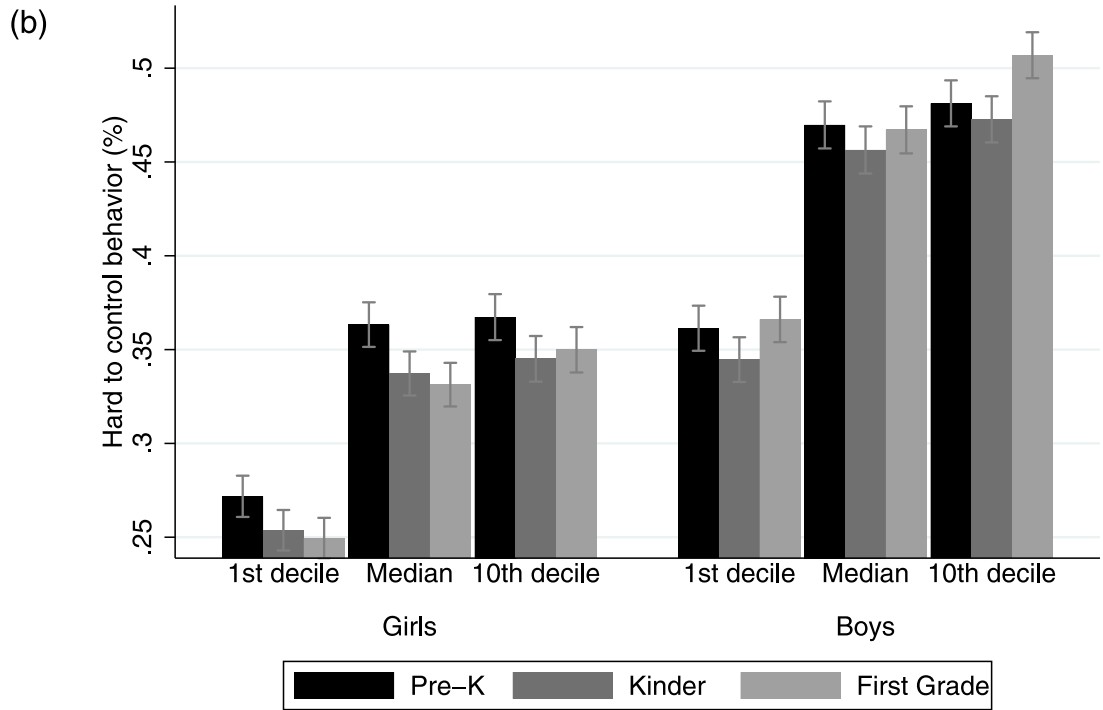

**Fig 1. Child nutritional health by school vulnerability.** (top) Obesity. (bottom) Behavior difficulties. Notes: Vulnerability deciles are constructed based on the school vulnerability index (IVE). Calculations based on the longitudinal matched JUNAEB data.

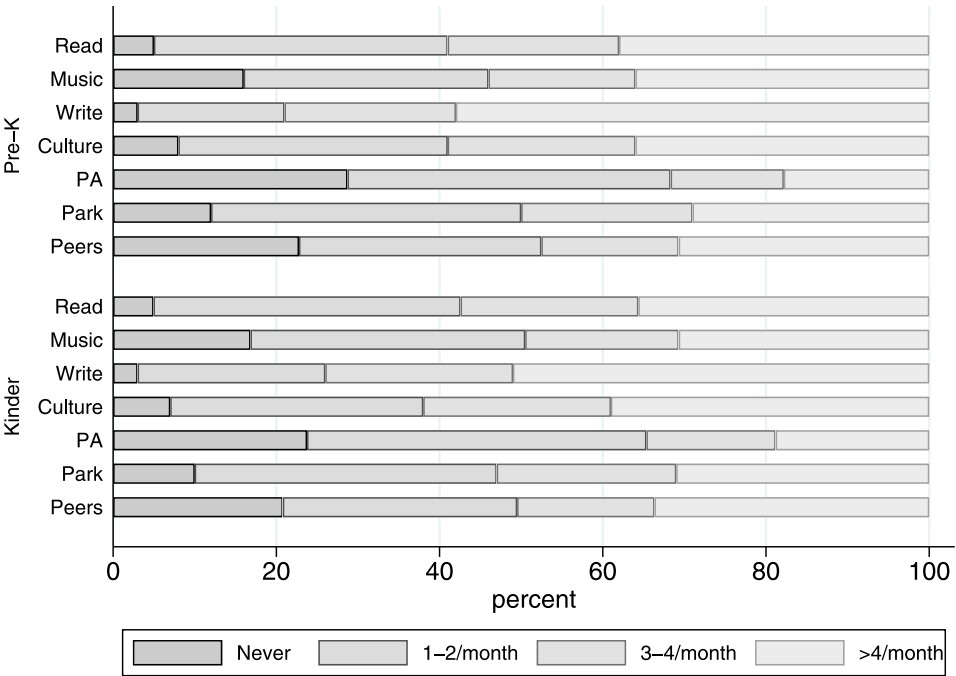

**Fig 2. Distribution of responses for time investments by grade and activity.** Culture indicates cultural activities, including visiting museums or watching a movie. Write includes writing or painting with the child. PA means physical activity, while peers refer to activities that involve similar-age children. Calculations based on the longitudinal matched JUNAEB data.

in the survey, based on the analytical sample. While there are remarkable differences between activities, on average only a third of all parents spend time in each activity at least once per week. Physical activity ans socialization outside school are the least frequent activities, while writing and reading are the most frequent time investments. Strikingly, more than 20% of caregivers declare to never engaged in physical activity or socialization with peers with their children in the last month.

## Child skill and health development technology

The standard framework for the technology of child development used in this study has been extensively discussed for cognition, socioemotional skills and health [13, 26–28]. I focus on nutritional health and socioemotional development since both are malleable and responsive to parental behaviors at preschool ages. In addition, theory suggests the potential complementarities between socioemotional development measures and nutritional status. In this study, nutritional status is measured by the BAZ, in its own scale, while socioemotional development is estimated from a measurement system. Based on previous work, I describe the dynamics of socioemotional development ($\theta_t$) and nutritional health ($H_t$) on a given period $t$, using a sequence of dynamic production functions that depend on parental behaviors (i.e., investments), initial conditions, household characteristics and the evolution of total factor productivity $A_t$ (which includes a random shock).

$$H_{t+1} = \delta_\theta ln\theta_t + \delta_H H_t + \delta_I lnI_t + lnA_{Ht} \tag{1}$$

$$ln\theta_{t+1} = \alpha_\theta ln\theta_t + \alpha_H H_t + \delta_I lnI_t + lnA_{\theta t} \tag{2}$$

Where $A_{Ht} = exp(\delta_{0t} + \delta_{Xt}X_t + e_t)$ and $A_{\theta t} = exp(\alpha_{0t} + \alpha_{Xt}X_t + v_t)$. Parental time investment (excluding basic care) is indexed by $I_t$, $P_t$ corresponds to parents' schooling attainment and $e_t$ and $v_t$ are idiosyncratic shocks. $X_t$ includes variables related to the family background (mother's age at birth, presence of a father figure, birth order and the number of siblings), and individual characteristics: birth weight, minority ethnic background, age, height-for-age z-scores (HAZ) and exclusive breastfeeding for six months. In the survey, respondents indicate whether the father figure is always present, sometimes or never, while also indicating the child's kinship. In 87% of cases, when a father figure is present, it corresponds to the biological father or the mother's partner. While family composition captures heterogeneity in parenting practices, individual data allows to account for heterogeneity in both growth patterns and early life investments.

## Parental time investments

Caregivers choose the allocation of time investments towards children's human capital based on individual preferences, time and resource constraints, and production technology [27, 29–31]. As noted in previous work [27], without explicit information on parental beliefs, estimating the structural model behind the dynamic optimization process impose strong assumptions contrary to recent evidence. In this analysis, the reduced form of the supply for time investments is log-linear, consistent with the approximation of the solution of a simple structural model.

$$lnI_t = \gamma_0 + \gamma_\theta ln\theta_t + \gamma_H H_t + \gamma_X X_t + \gamma_Z Z_t + u_t \qquad (3)$$

$X_t$ is enriched with additional variables that influence parental behavior, in particular, self-reported measures of social support in parenting and parental self-efficacy are included.

A critical issue to consider is the potential bias on the production function estimates that comes from endogenous parental time investments. Endogeneity can arise from unobserved inputs and correlated shocks between the supply of time investment and the production functions. The sign of bias, on average, depends on whether parents reinforce or substitute time allocation in response to unobserved inputs or shocks. Given a set of instruments $Z_t$, the control function approach is a natural strategy to test and account for potential endogeneity. If we assume linear conditional dependence between $e_t$, $v_t$, and $u_t$, we can include the estimated residual of the investment equation as an additional variable in the TFP. The estimated parameters of the residual allow for a direct test of endogeneity. The choice of the instruments must ensure that they are not correlated with the production function error term. From a theoretical perspective, variables included in the time and budget constraints are key candidates, such as observed relative price and quality of nearby schools, as well as access to physical inputs such as health services and green spaces. Recent experimental evidence from Chile shows that parents with preschool age children are likely to complement material investment goods (school choice) with parental time investments [32]. However, using current information on nearby school characteristics could be correlated with child outcomes due to peer effects [5]. As such, using prior information regarding nearby schools quality and prices, as well as access to other goods and services (e.g., parks, healthcare) can influence parents to substitute between leisure and time investments without impacting directly on skills and health, conditional on location choice. In the JUNAEB data, less than 4% all households move to a different commune between years.

Vector $Z_t$ includes the relative difference in standardized test scores (reading and math) for elementary school children (grades 2 and 4, respectively) in 2014, comparing the closest ten schools versus all the schools in the same commune (using the Euclidean distance). Also,

monthly school tuition in the year prior to the cohort data is included in bins ($2-$50, $50-$100, $100, or more). To incorporate monthly tuition as instruments, I set tuition-free schools as the base group and then create indicator variables per bin, which are set to one if there is at least one school with tuition cost in the corresponding price bin, for the ten closest schools.

## Latent factors and the measurement system

In the dataset, socioemotional development and time investments are partially captured by many categorical variables that characterize children's behavior. To avoid model selection over potential proxies and to address multiple sources of measurement error, I obtain latent factors from noisy proxies using a measurement system, that reduces dimensionality and accounts for measurement error bias [26, 33]. Explicitly, I define $\theta_t$ as the vector of all latent factors in the period $t$, where for a given $j$ factor, there are $k$ measurements. The measurement system then can be defined as:

$$M_{kt}^j = a_{kt}^j + \lambda_{kt}^j ln\theta_t^j + \eta_{kt}^j \tag{4}$$

$$\text{Factor Means}: E(ln\theta_t^j) = \mu_t^j \tag{5}$$

$$\text{Factor Covariance}: Var(\Theta) = \Omega_\theta \tag{6}$$

Where $a$ denotes factor intercepts, $\lambda$ indicates factor loadings, and $\eta$ are independent Gaussian measurement errors. Overall, this is a dedicated system, where each measure can only be associated with one factor. The structure of the measurement system was chosen based on exploratory factor analysis, or EFA for short (see S2 Appendix for an extensive discussion of the estimation of underlying factors from data).

Given that all measures are categorical, I follow the framework in [34] to account for longitudinal measurement invariance, in order to properly examine changes over time. The intuition is that repeated measures should capture the same latent factor (i.e., construct) in the same metric over time. If measurements for a given factor have $C$ response categories, latent measurement $M_{kt}^*$ is linked to the observed measurement $M_{kt}$ such that

$$M_{kt} = c \ \ \text{if} \ \ \tau_{c,jt} \leq M_{kt}^* < \tau_{c+1,jt} \tag{7}$$

Where $c = 0, 1, \ldots, C$ and $\tau_{c,jt}$ are threshold parameters to be estimated. In this case, I restrict thresholds for each measure to be the same over time, while allowing the variance of each measure to be unrestricted over time (i.e., threshold invariance model). This model guarantees that mean changes in the latent measurement over time are solely identified by changes in the latent factor. The latter condition is sufficient to characterize the dynamic nature of each latent factor from categorical indicators.

In addition, preliminary analysis of the data indicates a strong presence of parents' response styles in the behavioral observation of children's behavior (but not on parental time investments). Response styles can lead to extreme values across all measurements, affecting the estimated latent factors' quality. As such, following [35], I allow the intercepts to have a common (random) component across measurements and periods for each individual (parent) that is orthogonal to the underlying factors: $a_{ikt}^j = a_{it} + a_{kt}^j$. This random intercept captures the individual preference to report consistently lower (or higher) responses across all variables included in the measurement system, and it is assumed to be independent of the underlying measures of socioemotional development and parental investments (see S2 Appendix for more

details). With this additional structure, Eq (6) can be redefined as

$$M_{ikt}^{j*} = a_i + a_{kt}^j + \lambda_{kt}^j ln\theta_i^j t + \eta_{ikt}^j \tag{8}$$

The measurement system is identified if the means of log factors and measurement errors are set to zero, and the factor loading for the first measurement associated with each factor is fixed as one. In addition, to conduct valid inference, in each period, the latent factor is normalized to the same measurement, which determines scale In this case, all measurements have the same domain, since they are based on Likert-type scales or ordinal variables with equal numbers of potential responses.

## Estimation

The estimation is conducted in three steps. First, the measurement system's joint distribution is estimated from all observed measures and variables that enter the production functions and investment equations. The system is estimated by Means and Variance Adjusted Weighted Least Squares (WLSMV). The WLSMV estimator is robust to deviations from normality, common in ordinal data, such as Likert-type scales. Additionally, latent factors are estimated for each individual and period based on the linear prediction (Barlett scores). In the second step, time investment equations are estimated separately for each year, and the corresponding residuals are predicted. Finally, production functions are jointly estimated for each period, separately for boys and girls, using the control function approach. As in [23], both time investment supply and production functions are estimated at the sample means and at every decile of the distribution, to estimate the marginal productivity of investments through the empirical distribution of human capital. Standard errors are estimated using a nonparametric bootstrap procedure with 500 repetitions.

## Results

### Measurement system and latent factors

There is substantial evidence of response styles in the data, accounting for roughly 20% of the variance across measures (See S2 Appendix for more details). The estimated response styles correlate inversely with parental investments, mother's age, and parental schooling attainment, consistent with previous studies [36]. The associations suggest that more educated and older caregivers are more likely to identify children's behavioral difficulties.

Table 2 reports the variables allocated to each factor in the dedicated measurement system and the signal-to-noise ratios, i.e., the information content of each measure given the model specification. The formula for a given measure is:

$$s_j^{ln\theta_{kt}} = \frac{(\lambda_{kt}^j)^2 Var(ln\theta_{kt})}{(\lambda_{kt}^j)^2 Var(ln\theta_{kt}) + Var(\eta_{kt}^j)} \tag{9}$$

Questions regarding behavioral difficulties provide consistent information of a single latent factor over time, suggesting a single latent proxy of emotional stability. The assessment of parental time investments also indicates consistency across periods. Since all variables are categorical, each factor is scaled based on the empirical distribution of the latent measurements. However, given the longitudinal threshold invariance assumption, changes in the latent scale are associated with a given response category's probability of belonging. Moreover, results suggest that each measure's variance does not significantly vary over time, which allows to standardize the variance of the latent factor to the estimated variance in the first period. This

**Table 2. Signal-to-noise ratios.**

| | Pre-K | Kinder | First Grade |
|---|---|---|---|
| **Socioemotional skills** | | | |
| Hard to understand others | 39.8% | 39.2% | 42.9% |
| Hard to control behavior | 54.2% | 58.5% | 62.9% |
| Hard to get along with peers | 59.3% | 60.2% | 64.6% |
| **Parental time investment** | | | |
| Read to child | 45.4% | 41.7% | |
| Plays music | 33.1% | 34.7% | |
| Writes or paints | 45.0% | 46.7% | |
| Cultural activities | 32.8% | 34.4% | |
| Physical activity | 52.6% | 54.4% | |
| Goes to park | 53.9% | 55.7% | |
| Socializes with peers | 27.4% | 28.8% | |

Table notes Percentages represent the fraction of total variance of each measure associated with the estimated factor.

standardization permits the prediction of each log-factor in the metric of a standardized z-score, in order to be comparable to the measure of nutritional health.

Fig 3 shows the mean levels of estimated parental time investments by year and sex at different deciles of SED and BAZ. Children at the top of the distribution of socioemotional development receive roughly 0.5 SD higher parental time inputs, compared with children in the first decile. Boys receive slightly higher parental time allocation, however differences are not statistically significant. In terms of BAZ, there are no mayor differences between time inputs by gender or nutritional status, except for girls with high BAZ in Pre-K.

## Determinants of parental time investments

Table 3 reports the estimated parameters for the time investment equations, and the estimated standard errors via bootstrap (clustered the commune level). All variables are expressed in logs except for categorical indicators. The estimates reflect the first stage where parental time inputs are instrumented using school and neighborhood characteristics.

Parents seem to reinforce time investments to the observed socioemotional development, while BAZ is not relevant, in magnitude. Regarding children's characteristics, while age, sex, and HAZ are not quite relevant, parents invest more time with children who are first born and those with fewer siblings. While parental education contribute little to differences in time investments, the permanent presence of a father figure, parental self-efficacy, social support for parenting and participation in social organizations contribute greatly. Strikingly, parents that have low self-efficacy (i.e. indicate that parenting is hard) spend 11 percent points less time, on average. Interestingly, while most determinants of parental time investments remain stable between grades, the salience of socioemotional development increases significantly between Pre-Kinder and Kindergarten. Similarly, the constant presence of a father figure becomes more relevant for older children. This is quite relevant as one-third of all children live without a father, and 7% have no father figure in the household by the time they enter elementary school.

In terms of instruments, joint significance of school prices and quality is strong in both periods. Moreover, using the Montiel-Pflueger test, there is no weak instruments problem [37]. Estimates suggests that caregivers are likely to increase parental time allocation in response to the perceived quality and price of schools and amenities in the relevant

(a)

(b)

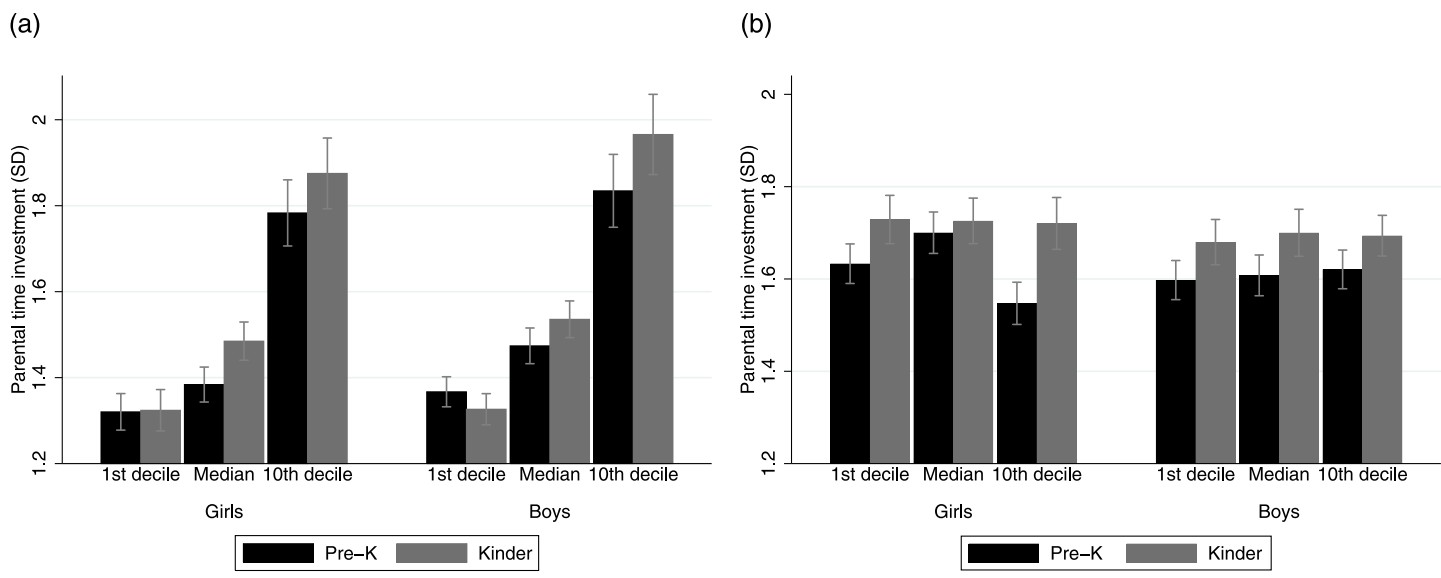

(c)

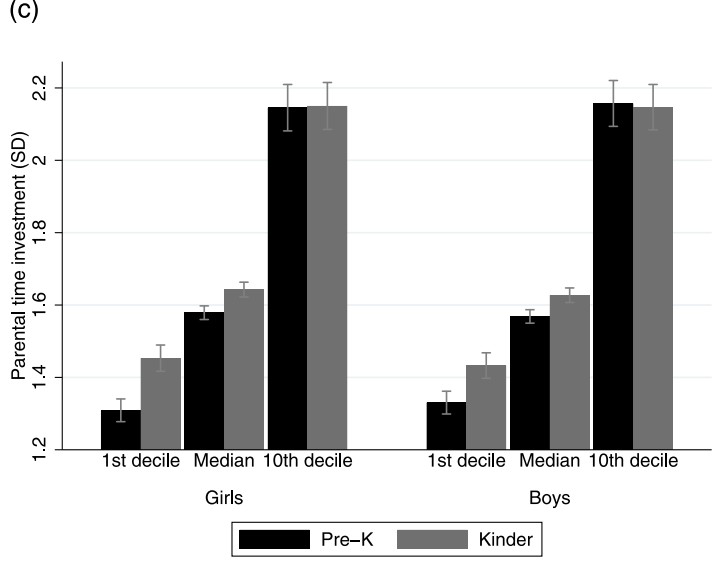

**Fig 3. Parental time investment by child skills and mother education.** (top) Socioemotional development. (middle) BMI for age z-scores. (bottom) Mother education (years). Notes: Deciles are constructed based on the relevant variable in each period. Calculations based on the longitudinal matched JUNAEB data. Latent scales are constructed so log means are zero.

neighborhood. Moreover, given the evidence of limited inter-generational mobility in Chile [38], these results confirm that information on material investments and local resources is relevant for parental time allocation.

## Production function estimates

Table 4 shows the estimates of the production functions for each year and sex, accounting for endogeneity on parental time allocation. Parental investment elasticity to human capital is roughly 0.1 and rather constant between grades. The persistence in SED is large and increasing

**Table 3. Parental time investments (in logs).**

|  |  | Pre-Kinder |  | Kindergarten |  |
|---|---|---|---|---|---|
| Skills (log) |  | **0.064** | *0.003* | **0.101** | *0.003* |
| BAZ |  | 0.001 | *0.001* | **0.004** | *0.001* |
| Age (log) |  | 0.000 | *0.034* | 0.000 | *0.034* |
| HAZ |  | **0.006** | *0.003* | **0.008** | *0.001* |
| Gender (male = 1) |  | **0.014** | *0.003* | **0.011** | *0.003* |
| First born |  | **0.067** | *0.004* | **0.055** | *0.004* |
| Exclusive breastfeeding |  | **0.050** | *0.003* | **0.040** | *0.003* |
| Number of siblings |  | **-0.048** | *0.002* | **-0.055** | *0.002* |
| Caretakers (number) |  | **0.032** | *0.002* | **0.033** | *0.002* |
| Ethnic background = 1 |  | **-0.060** | *0.007* | **-0.045** | *0.007* |
| Household in urban area = 1 |  | **0.064** | *0.007* | **0.054** | *0.007* |
| Mother age at birth (log) |  | **-0.061** | *0.009* | **-0.094** | *0.010* |
| Mother education (log years) |  | **0.028** | *0.004* | **0.020** | *0.003* |
| Father education (log years) |  | **0.030** | *0.002* | **0.019** | *0.002* |
| Mother in salary work = 1 |  | -0.002 | *0.004* | -0.012 | *0.006* |
| Father in salary work = 1 |  | -0.001 | *0.005* | -0.001 | *0.005* |
| Mother self-employed = 1 |  | **0.021** | *0.004* | **0.034** | *0.004* |
| Father self-employed = 1 |  | **0.019** | *0.005* | **0.044** | *0.005* |
| Father figure present (Never) |  |  |  |  |  |
|  | Sometimes | **0.015** | *0.007* | **0.023** | *0.008* |
|  | Always | **0.121** | *0.007* | **0.142** | *0.008* |
| Pareting this child is (Easy) |  |  |  |  |  |
|  | Not easy nor hard | **-0.057** | 0.003 | **-0.050** | 0.003 |
|  | Hard | **-0.128** | 0.007 | **-0.134** | 0.009 |
| Pareting support (Always) |  |  |  |  |  |
|  | Sometimes | **-0.065** | 0.003 | **-0.065** | 0.003 |
|  | Never | **-0.070** | 0.006 | **-0.075** | 0.006 |
| Participation in social org. |  | **0.128** | *0.003* | **0.126** | *0.003* |
| Home close to recreation area |  | **0.190** | *0.004* | **0.201** | *0.005* |
| Home close to public services |  | **0.059** | *0.006* | **0.069** | *0.006* |
| School tuition (monthly) |  |  |  |  |  |
|  | $2 to $50 | **0.065** | *0.009* | **0.070** | *0.008* |
|  | $50 to $100 | **0.093** | *0.008* | **0.080** | *0.008* |
|  | $100 or more | **0.129** | *0.009* | **0.070** | *0.009* |
| School math z-score (grade 4) |  | **0.014** | *0.003* | **0.013** | *0.003* |
| School reading z-score (grade 2) |  | 0.005 | *0.003* | 0.006 | *0.003* |
| Instruments F-stat (p-value) |  | **67.46** | *0.000* | **70.56** | *0.000* |
| Montiel-Pflueger test (p-value) |  | **180.32** | *0.000* | **142.23** | *0.000* |
| N |  | 86,139 |  | 92,081 |  |

Table notes Significant values in bold (p<0.1). In the school tuition categories, the excluded group is public, tuition-free schools. Based on information from the Ministry of Education, no schools have tuition prices between 0 − $2 dollars. Standard errors in italics.

from Kindergarten to First grade, consistent with previous evidence for non-cognitive abilities in the literature [26]. Nutritional health and socioemotional development are weak complements; children with higher BAZ have lower socioemotional development in the next period; however, on average, the magnitude is quite small. While parental education is significantly

**Table 4. Socioemotional development and nutritional health production functions.**

| | Socioemotional (t+1) | | | | BAZ (t+1) | | | |
|---|---|---|---|---|---|---|---|---|
| | Kindergarten | | First grade | | Kindergarten | | First grade | |
| | **Boys** | **Girls** | **Boys** | **Girls** | **Boys** | **Girls** | **Boys** | **Girls** |
| Investment | **0.11** | **0.08** | **0.10** | **0.08** | **-0.15** | **-0.10** | **-0.07** | **-0.08** |
| | 0.02 | 0.02 | 0.03 | 0.02 | 0.08 | 0.04 | 0.04 | 0.05 |
| BAZ | **-0.01** | 0.00 | **-0.01** | **-0.01** | **0.40** | **0.40** | **0.48** | **0.49** |
| | 0.00 | 0.00 | 0.00 | 0.00 | 0.01 | 0.01 | 0.01 | 0.01 |
| Socioemotional | **0.68** | **0.67** | **0.79** | **0.77** | **-0.02** | -0.01 | **-0.02** | **-0.02** |
| | 0.01 | 0.01 | 0.01 | 0.01 | 0.01 | 0.01 | 0.01 | 0.01 |
| Mother education (log years) | **0.01** | 0.00 | **0.02** | **0.02** | -0.01 | -0.02 | -0.03 | -0.02 |
| | 0.00 | 0.00 | 0.01 | 0.01 | 0.01 | 0.01 | 0.01 | 0.01 |
| Father education (log years) | **0.01** | 0.01 | **0.01** | 0.00 | 0.00 | -0.01 | -0.01 | -0.01 |
| | 0.00 | 0.00 | 0.00 | 0.00 | 0.01 | 0.01 | 0.01 | 0.01 |
| Mother's age | **0.12** | **0.12** | **0.15** | **0.13** | **0.06** | **0.07** | 0.01 | **0.07** |
| | 0.01 | 0.02 | 0.01 | 0.01 | 0.03 | 0.03 | 0.03 | 0.03 |
| Father figure (Never) | | | | | | | | |
| Sometimes | 0.00 | 0.00 | 0.00 | 0.00 | 0.02 | -0.03 | -0.01 | 0.04 |
| | 0.01 | 0.01 | 0.01 | 0.01 | 0.03 | 0.03 | 0.02 | 0.01 |
| Always | **0.07** | **0.06** | **0.13** | **0.10** | -0.01 | -0.01 | -0.01 | **0.04** |
| | 0.01 | 0.01 | 0.02 | 0.01 | 0.03 | 0.04 | 0.03 | 0.02 |
| Age (log months) | 0.03 | 0.00 | **0.20** | **0.14** | **0.27** | **0.30** | **0.58** | **0.35** |
| | 0.02 | 0.02 | 0.05 | 0.04 | 0.09 | 0.08 | 0.03 | 0.07 |
| HAZ | 0.00 | **0.00** | 0.00 | 0.00 | **0.20** | **0.20** | **0.26** | **0.25** |
| | 0.00 | 0.00 | 0.00 | 0.00 | 0.01 | 0.01 | 0.01 | 0.01 |
| Weight at birth (log kg) | 0.01 | -0.02 | 0.01 | 0.01 | **0.40** | **0.35** | **0.35** | **0.36** |
| | 0.01 | 0.01 | 0.01 | 0.02 | 0.04 | 0.03 | 0.03 | 0.03 |
| Exclusive breastfeeding | **0.01** | 0.00 | **0.01** | **0.01** | **0.03** | 0.01 | **0.03** | **0.02** |
| | 0.00 | 0.00 | 0.01 | 0.01 | 0.01 | 0.01 | 0.01 | 0.01 |
| First born = 1 | **-0.01** | 0.00 | **0.02** | **0.02** | 0.01 | -0.01 | -0.01 | 0.01 |
| | 0.00 | 0.00 | 0.01 | 0.00 | 0.01 | 0.01 | 0.01 | 0.01 |
| Number of siblings | **-0.01** | **-0.01** | **-0.02** | **-0.01** | **-0.05** | **-0.04** | **-0.04** | **-0.03** |
| | 0.00 | 0.00 | 0.00 | 0.00 | 0.01 | 0.01 | 0.01 | 0.01 |
| Ethnic background = 1 | 0.01 | 0.01 | **0.02** | **-0.02** | **0.11** | **0.04** | **0.10** | **0.07** |
| | 0.01 | 0.01 | 0.01 | 0.01 | 0.02 | 0.02 | 0.02 | 0.02 |
| Lives in urban area = 1 | 0.01 | 0.01 | **0.02** | **-0.02** | -0.02 | -0.02 | -0.02 | -0.02 |
| | 0.01 | 0.01 | 0.01 | 0.01 | 0.02 | 0.02 | 0.02 | 0.02 |
| Investment Res. | **-0.09** | **-0.06** | **-0.01** | **-0.06** | **0.17** | **0.08** | **0.08** | **0.07** |
| | 0.02 | 0.02 | 0.03 | 0.03 | 0.06 | 0.02 | 0.04 | 0.03 |
| Adjusted R-squared | 0.48 | 0.46 | 0.49 | 0.47 | 0.19 | 0.21 | 0.31 | 0.34 |
| N | 45,661 | 46,680 | 45,522 | 48,572 | 45,661 | 46,680 | 45,522 | 48,572 |

Table notes Significant values in bold (p<0.1). Mother's and Father's education, as well as mother's age are measured in log-years. Standard errors in italics.

related to skill production, the magnitude is negligible. However, the mother's age is strongly related to higher socioemotional development. The constant presence of a father figure has a remarkable effect on socioemotional development, after accounting for time investments, which might suggest an unobserved channel not captured in the time investments (e.g. role modeling). Given the longitudinal balance of the analytical sample, estimates suggests that,

everything else constant, older children have higher SED but also increased BAZ. Relatively older children are more exposed to socialization, which can facilitate skill accumulation, but it can also reinforce unhealthy eating behaviors.

For nutritional health, parental time investments have a significant effect on BAZ, but its importance decreases over time, as persistence increases. There is weak evidence of complementarity between dimensions of human capital, increased socioemotional development leads to lower BAZ in the next period. As expected, after accounting for seasonal patterns, age, and HAZ explain a significant part of the BAZ in a given year. Taller and older children within the cohort are more likely to be overweight and obese, consistent with previous longitudinal evidence [39]. Weight at birth also contributes substantially, in line with emerging evidence on the importance of managing weight at birth. [40] show that Turkish children with a weight higher than 3.8 kilos are at greater risk of being overweight or obese during early childhood, after controlling for feeding practices and parental characteristics. Finally, there is strong evidence of endogeneity in both socioemotional development and HAZ. Assuming exogeneity of parental investments leads to lower estimates of their impact of human capital development. The latter suggests that, conditional on instruments for material investments and resources, there is a negative correlation between time allocation and unobserved inputs (S3 Appendix shows the production functions' estimates without using the control function approach). As cognition is plausibly the most important unobserved input, estimates suggests that parents allocate less time to children with higher cognitive skills.

## Distributional effects of parental time investments

Estimates on potential effectiveness of parental time investments by decile is shown in Fig 4 (estimates on input complementarities are presented in S4 Appendix). Evidence suggests that the marginal productivity of time investments on socioemotional development decreases for children at the top of the SED distribution. Children with lower socioemotional development at baseline can benefit from additional time, up to 0.4 SD. However, increasing time investments could harm children at the top of the distribution. Is possible that *intensive parenting*, i.e. parents introducing excessive structured activities leading to overcrowding, could decrease (or at least not improve) developmental outcomes [41]. Unfortunately, without observing quality of time investments, is not possible to determine the underlying mechanisms for negative returns to time investments among children with high SED.

The results for BAZ are also remarkably impressive. The impact of time investments is inversely related to BAZ at baseline, and the impact could be up to 0.8 SD reduction among severely obese children. While the effects are higher for boys than girls in Kindergarten, the effects are quite similar for both sexes in first grade, being only significant for overweight and obese children (top of the BAZ distribution). Evidence from labor studies in the United States and other developed countries indicates that lower time in-home childcare due to labor supply variation can substantially increase children's obesity risk at school age [8, 42, 43]. Given the diverse tasks included in the time investments, two mechanisms that could explain the results. First, at least two of the tasks included in the measures involve some form of physical (recreational) activity, which directly impacts BAZ, all else constant. Secondly, as parental time investments are highly correlated with socioeconomic conditions, is expected that stimulation activities are correlated with diet quality at home [44, 45].

Overall, evidence suggest that children experiencing developmental gaps are those who would largely benefit from additional parental time allocation. However it is unclear to which extent there is room for trade-offs with leisure, consumption and other time costs at the household level (e.g., time allocated to child basic care or transport if recreation areas are far from

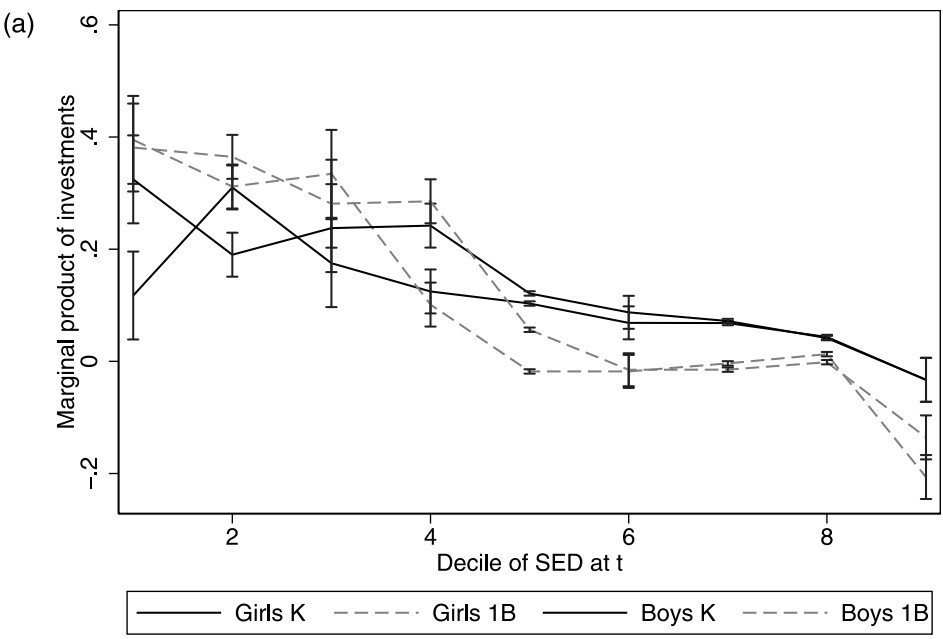

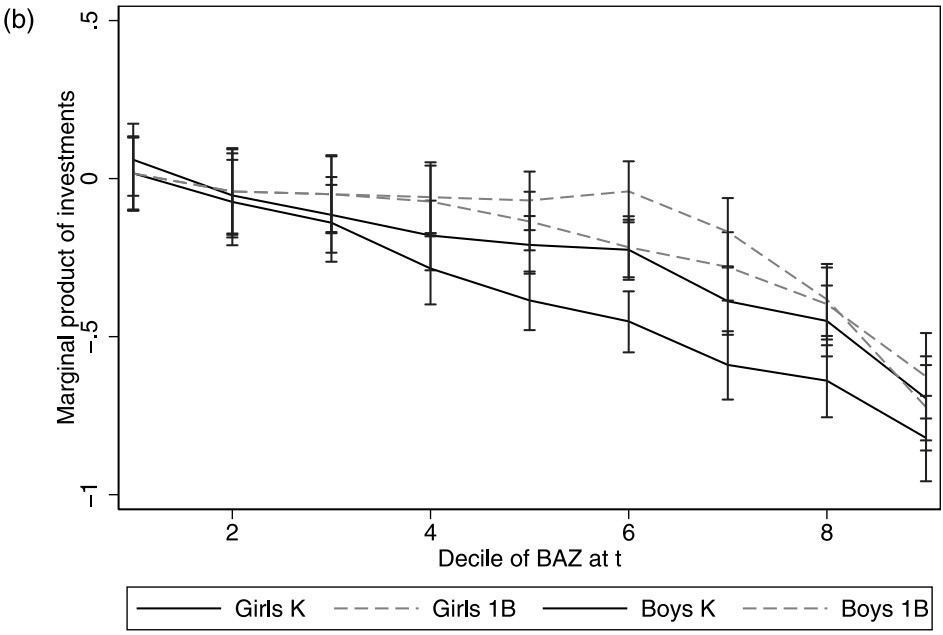

**Fig 4. Marginal product of parental time investments.** (top) Socioemotional development. (bottom) BMI for age z-scores. Notes: Vulnerability deciles are constructed based on the school vulnerability index (IVE). Calculations based on the longitudinal matched JUNAEB data. Latent scales are constructed, so log means are zero.

home, cooking time versus eating pre-made meals). In addition, consistent with evidence from parenting interventions, self-efficacy and social support play a major role on parental time investments [19]. After accounting for family composition, child human capital, and resources, caregivers invest 12% less time if they perceive parenting as *hard*, compared to those that consider it *easy*. Similarly, parenting support from a stable father figure presence, as well

as from a social support network, are a key to meaningfully increase time investments. While the complementary between socioemotional development and BAZ is low, the potential effects of interventions boosting parental time investments are quite promising. In particular, given that the coexistence of excess weight and limited behavioral control socioemotional development among vulnerable students. In the analytical sample, the obesity prevalence among children at the bottom of the SED distribution is 45% higher compared to children with high socioemotional development.

## Discussion

Recent evidence suggests that the quality and quantity of time investments devoted by caregivers have a significant effect on health and socioemotional development in the first years of life [27, 46]. This study presents evidence from a complete cohort of all students starting Pre-Kinder in public schools in 2015, identifying the potential that parental time allocation has on both obesity risk and socioemotional development. First, following the framework discussed in [12, 26], I estimate measures of parental time investments and developmental socioemotional development using a measurement system that accounts for the categorical nature of the data and extreme response styles. Secondly, using the latent factors, I estimate the parent's time investment schedule and obtain the residuals in order to account for endogeneity in the estimation of the production functions.

Results from the investment equations reveal that caregivers' time allocation respond to children's socioemotional development but not to body mass index z-scores. Social support and self-efficacy are important determinants of variation in time investments. Moreover, access to public goods and the price and quality of nearby schools contribute to explain parental behavior. The latter suggests potential complementarities between time and material investments. Results also indicate that vulnerable households are bounded by time and resource constraints in order to provide stimulation and nutrition at preschool age optimally. Still, caregivers could benefit greatly from behavior change interventions aimed to provide self-efficacy and support networks. In this context, extending universal coverage to successful, ongoing parenting training programs provided through the health and education systems, such as *Nadie es Perfecto* and *Habilidades para la Vida*, could substantially benefit the development of vulnerable young children. The estimates from production functions that assume investments as exogenous exhibit a downward bias, most likely due to a negative correlation between unobserved inputs, particularly cognition, and parental time investments. Evidence from other developed countries suggests that parents compensate with additional time to children with low cognitive outcomes, which supports the idea of a downward bias on the effectiveness of parental time investments when cognition is omitted form the production functions [6].

Regarding the production functions, time investments have a significant impact on both future SED and BAZ. The effects are quite substantial for vulnerable children, consistent with experimental evidence from randomized interventions. However, results also offer a word of caution: measures of time investments could also capture how unresponsive or intensive parenting could harm children's socioemotional development at the top of the distribution. The effects of additional parenting time on body mass reduction are quite substantial. In perspective, recent evidence of the structural policies targeted to the food environment in Chile shows that changes in the total energy intake are significant but not meaningful to significantly impact body mass indices among children (only a few calories per day). In contrast, increasing parental activities from 1–2 to 3–4 times per month in a year can reduce BAZ up to 0.8 SD among severely obese children. A recent evaluation of the

Chilean School Meals Program shows that is conducive to a (local) reduction on BAZ of 0.3 SD among obese girls in first grade [47]. Given that more than 20% of caregivers do not engage in physical activities or peer socialization with their children, there is substantial scope to shape policies to favor not only access to recreational areas and information, but also promote self-efficacy and social support through interpersonal communication through social organizations. In terms of dynamic complementarities, evidence of the prospective role of SED on child outcomes provides partial evidence that persistent high BAZ among Chilean children is most likely due to poor food environments rather than unhealthy behaviors, which explains the weak correlation between socioemotional development and nutritional health.

There are two important limitations to the analyses presented in this paper. First, since JUNAEB administrative dataset follows children attending public funded schools, we cannot infer the role of parental time investments for rich households with children attending private schools (roughly ten percent of all children). Marginal returns to parental time investments could be enhanced by school quality, providing additional incentive for parents to increase their time allocation. However, evidence from the instruments used in the parental time investment equations suggests that parents respond to school quality (in the available data), thus the relationship is likely to be monotonic for private schools with higher quality. Secondly, while the administrative data is quite rich, we do not have information regarding income or wages, which are key to understand the scope for trade-offs between labor supply, material and time investments. As noted by [48], income effects could be quite significant, albeit there is no evidence that income could affect the productivity of time investments (except via indirect effects on material investments, such as school choice).

## Conclusion

Child obesity is a global epidemic, steadily increasing in the last decades. Developing and developed countries are concentrating their efforts on enacting strict regulations to shape their food systems in order to mitigate the obesity epidemic, with limited success [49, 50]. However, significant evidence from observational studies, RCTs, and large interventions indicates that providing support and training to parents can substantially affect the quality and quantity of material and time investments towards children's development and optimal nutrition at preschool and beyond. In the light of the evidence presented in this study, such programs can be extremely successful to prevent obesity among children in the short term, and also to avoid excess weight over the life-cycle by fostering socioemotional development that promotes the adoption of healthy behaviors.

## Supporting information

**S1 Appendix. National Board of School Aid and Scholarships.**
(PDF)

**S2 Appendix. Socioemotional development and parental investments.**
(PDF)

**S3 Appendix. Robustness checks.**
(PDF)

**S4 Appendix. Heterogeneity on input complementarities.**
(PDF)

## Acknowledgments

This work includes substantial effort from JUNAEB, particularly Mariana Lira, who provided insights into the data collection process. The contents and opinions in this article are solely the personal views of the author. I affirm that all remaining errors are my own.

## Author Contributions

**Conceptualization:** Juan Carlos Caro.

**Data curation:** Juan Carlos Caro.

**Formal analysis:** Juan Carlos Caro.

**Methodology:** Juan Carlos Caro.

**Validation:** Juan Carlos Caro.

**Writing – original draft:** Juan Carlos Caro.

**Writing – review & editing:** Juan Carlos Caro.

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
