## [Decision Letter · Decision Letter 0]

27 Mar 2023

PONE-D-23-03208Distributional effects of parental time investments on children's socioemotional skills and nutritional healthPLOS ONE

Dear Dr. Caro,

Thank you for submitting your manuscript to PLOS ONE. After careful consideration, we feel that it has merit but does not fully meet PLOS ONE’s publication criteria as it currently stands. Therefore, we invite you to submit a revised version of the manuscript that addresses the points raised during the review process.

We look forward to receiving your revised manuscript.

Kind regards,

José Alberto Molina

Academic Editor

PLOS ONE

Journal Requirements:

2. Please update your submission to use the PLOS LaTeX template. The template and more information on our requirements for LaTeX submissions can be found at " ext-link-type="uri" xlink:type="simple">http://journals.plos.org/plosone/s/latex."

"This work was partially funded by National Agency for

Research and Development of Chile (ANID) through grant PAI/INDUSTRIA

79090016. The contents and opinions in this article are solely the personal

views of the author. I affirm that all remaining errors are our own."

Reviewers' comments:

Reviewer's Responses to Questions

**Comments to the Author**

1. Is the manuscript technically sound, and do the data support the conclusions?

Reviewer #1: Yes

Reviewer #2: Yes

2. Has the statistical analysis been performed appropriately and rigorously? 

Reviewer #1: Yes

Reviewer #2: Yes

3. Have the authors made all data underlying the findings in their manuscript fully available?

Reviewer #1: No

Reviewer #2: Yes

4. Is the manuscript presented in an intelligible fashion and written in standard English?

Reviewer #1: Yes

Reviewer #2: Yes

5. Review Comments to the Author

Reviewer #1: The manuscript studies a very interesting topic, using an adequate methodology. The paper is well written and polished. The contribution and policy implications are clearly stated. My main suggestion consists on elaborating and explaining more the prior literature – without collapsing so many references. Further, it would be good to map for which countries there is prior evidence (e.g. is there similar evidence for other Latin American countries?).

Minor comments: there are a few typos, such as “An critical issue” (page 8).

Reviewer #2: Distributional effects of parental time investments on children's socioemotional skills and nutritional health

PONE-D-23-03208

Reviewer report

Comments to author

First, I would like to point out to the author that I found their paper quite interesting and enjoyed reading it.

However, I would like to give some recommendations that may contribute to improve this work.

1. I have missed an explanation of why the author has selected this country and not another one for this paper.

2. I consider that the data used for this paper are correct, but it should be mentioned as a limitation of this paper that there is no information on private schools. Hofflinger et al. (2020) show in Chile that a key assumption of school choice and competition policies is that parents' most important (if not only) priority in choosing a school is its quality. Chile is a country with a national system of school choice and competition and in this sense the authors in their study find that parents who choose a school prioritize its proximity, its quality and whether it provides religious education. In their results they show that the probability that parents prioritize proximity is higher for parents of low socioeconomic status, while the probability that they prioritize religious education and quality is higher for parents of high socioeconomic status. Their findings show that only advantaged families choose schools based on their quality and, therefore, school choice and competition policies may offer limited benefit for disadvantaged students, possibly maintaining or reinforcing socioeconomic segregation in the educational system.

It would be necessary to mention this data limitation in the paper.

3. It would be important to mention and go a little deeper into the paper, that there are different types of child care. The literature shows that it is important to classify the type of child care, and some authors classify it into three types: basic, educational, and supervisory child care (Guryan et al., 2008; Gimenez-Nadal and Molina, 2012; Campaña et al., 2017). Within these three activities mentioned above, the activities aimed at increasing the human capital of children are those found in educational child care. And regarding differences between fathers and mothers: Fathers prefer childcare activities that are more rewarding, such as playing with their children or helping them with homework (Craig, 2006a; Darling-Fisher Tiedje, 1990; Giménez-Nadal Molina, 2013; Grossman, Pollack, Golding, 1988; Kahneman Krueger, 2006; Kahneman, Krueger, Schkade, 2004).

4. It is important to delve a little deeper into the following: The existing literature clearly shows that parents with high levels of education tend to spend more time with their children, compared with parents with low levels of education (Bianchi, Cohen, Raley, Nomaguchi, 2004; Campaña et al., 2017; Craig, 2006b; Gimenez-Nadal Molina, 2013; Guryan, Hurst, Kearney, 2008; Hofferth, 2001; Marsiglio, 1991; Sayer, Bianchi, Robinson, 2004; Sayer et al., 2004) partly due to the fact that more educated parents recognise and acknowledge the importance of time investments in their children (Kalenkoski Foster, 2008; Marsiglio, 1991; Sayer et al., 2004; Sevilla Borra, 2015). Better educated parents spend more time in educational childcare activities, including reading, playing, and helping with homework (Gimenez-Nadal Molina, 2013; Hill Stafford, 1985; Kalenkoski Foster, 2008; Sayer et al., 2004), that promote the development of human capital of children (Brooks-Gunn et al., 2002).

5. It is feasible to include in the analysis a variable indicating the presence or number of other household members (other than the father, mother or younger siblings of the children analyzed). Delgado and Canabal (2006) and Fuller, Holloway, and Liang (1996) show that members of Hispanic families allocate their time according to their strong family orientation (focusing on the family group), and parents tend to use other family members to care for their children.

6. Why are individuals' wages and non-labor income not taken into account in the explanatory variables? Research has shown that higher wages lead to a better position at home as it increases the bargaining power within the couple (Bourguignon, Browning, Chiappori, 2009; Chiappori, 1988, 1992; Lundberg Pollak, 1993), and this higher bargaining power may be used to take responsibility for more rewarding childcare activities, such as playing with or reading to children. Non-labour income may also affect the time parents devote to childcare. Kalenkoski, Ribar, and Stratton (2005) show that mothers reduce the time devoted to active childcare when household income increases.

7. It would be feasible to include a dummy variable of whether the children analyzed live in an urban or rural area. The female labor participation rate is lower in rural areas than in urban areas. In addition, there are other control variables, such as indigenous population and family structure, which can vary considerably from one area to another.

8. Regarding mothers' work, it is feasible to know whether they are self-employed or salaried. The literature shows a positive relationship between self-employment and time spent on childcare (Conelly 1992; Edwards and Field-Hendrey 1996; Caputo and Dolinsky 1998; Boden 1999; Gimenez et al., 2013; Campaña et al., 2020).

6. PLOS authors have the option to publish the peer review history of their article (what does this mean?). If published, this will include your full peer review and any attached files.

Reviewer #1: No

Reviewer #2: No

---

## [Author Response · Author response to Decision Letter 0]

25 May 2023

Reviewer #1: The manuscript studies a very interesting topic, using an adequate methodology. The paper is well written and polished. The contribution and policy implications are clearly stated. My main suggestion consists on elaborating and explaining more the prior literature – without collapsing so many references. Further, it would be good to map for which countries there is prior evidence (e.g. is there similar evidence for other Latin American countries?).

R: I appreciate the overall review of the paper. I acknowledge the need for further description of the prior literature, which has been extended in the new version of the manuscript. Despite the limited evidence in distributional effects to date, I extended the discussion regarding the heterogeneity (based on observables) of the link between parental time investments and child development. As noted in the manuscript, to the best of our knowledge, there is no evidence on heterogeneity of parental time investments on child development for any country, in contrast with material and health investments, as presented in recent work:

Asfaw, A.A. (2018). The distributional effect of investment in early childhood

nutrition: A panel quantile approach. World Development, 110 , 63–74. 

Attanasio, O., Meghir, C., Nix, E. (2020). Human capital development and

parental investment in india. The Review of Economic Studies, 87 (6),

2511–2541.

Lee, S.Y., Rodgers, J., Kim, R., Subramanian, S. (2022). Distributional effects

on children’s cognitive and social-emotional outcomes in the head start

impact study: A quantile regression approach. SSM-Population Health,

18 , 101108.

Minor comments: there are a few typos, such as “An critical issue” (page 8).

R: I apologize for the typos, which are corrected in the new version of the manuscript.

Reviewer #2: Distributional effects of parental time investments on children's socioemotional skills and nutritional health

First, I would like to point out to the author that I found their paper quite interesting and enjoyed reading it. However, I would like to give some recommendations that may contribute to improve this work.

1. I have missed an explanation of why the author has selected this country and not another one for this paper.

R: Thank you for this comment. There are two key factors that makes the data from Chile valuable for this analysis. First, Chile is a high-income country that experienced the increase in childhood obesity prevalence that is common in similar countries in the nutrition transition path towards ultra-processed foods. Secondly, the administrative data used in the analysis is unique, and to my knowledge, there is no other country with a set of longitudinal data as comprehensive as the one provided by JUNAEB. 

2. I consider that the data used for this paper are correct, but it should be mentioned as a limitation of this paper that there is no information on private schools. Hofflinger et al. (2020) show in Chile that a key assumption of school choice and competition policies is that parents' most important (if not only) priority in choosing a school is its quality. Chile is a country with a national system of school choice and competition and in this sense the authors in their study find that parents who choose a school prioritize its proximity, its quality and whether it provides religious education. In their results they show that the probability that parents prioritize proximity is higher for parents of low socioeconomic status, while the probability that they prioritize religious education and quality is higher for parents of high socioeconomic status. Their findings show that only advantaged families choose schools based on their quality and, therefore, school choice and competition policies may offer limited benefit for disadvantaged students, possibly maintaining or reinforcing socioeconomic segregation in the educational system. It would be necessary to mention this data limitation in the paper.

R: Thank you for the helpful remark. Indeed, we adapted the manuscript to emphasize that the effects are related to parents with children attending public or public-funded schools. In that sense, we cannot extrapolate regarding the impact of time allocation on parental investments among wealthy households, which represent around 10% of the total school enrollment on a given year. 

3. It would be important to mention and go a little deeper into the paper, that there are different types of child care. The literature shows that it is important to classify the type of child care, and some authors classify it into three types: basic, educational, and supervisory child care (Guryan et al., 2008; Gimenez-Nadal and Molina, 2012; Campaña et al., 2017). Within these three activities mentioned above, the activities aimed at increasing the human capital of children are those found in educational child care. And regarding differences between fathers and mothers: Fathers prefer childcare activities that are more rewarding, such as playing with their children or helping them with homework (Craig, 2006a; Darling-Fisher Tiedje, 1990; Giménez-Nadal Molina, 2013; Grossman, Pollack, Golding, 1988; Kahneman Krueger, 2006; Kahneman, Krueger, Schkade, 2004).

R: This is quite an interesting point, and we included a further discussion in the document, to the extent that the data permits. Based on the questions in the survey, our findings are related to time allocated in stimulation activities, which could be considered as educational child care. In this paper we are also neutral regarding the caregiver engaging in these activities with the children, since the survey respondent could be one or both parents, or another person who is appointed as primary caregiver. As such, we cannot address gender preferences in time investments. We acknowledge this in the limitations section.

4. It is important to delve a little deeper into the following: The existing literature clearly shows that parents with high levels of education tend to spend more time with their children, compared with parents with low levels of education (Bianchi, Cohen, Raley, Nomaguchi, 2004; Campaña et al., 2017; Craig, 2006b; Gimenez-Nadal Molina, 2013; Guryan, Hurst, Kearney, 2008; Hofferth, 2001; Marsiglio, 1991; Sayer, Bianchi, Robinson, 2004; Sayer et al., 2004) partly due to the fact that more educated parents recognise and acknowledge the importance of time investments in their children (Kalenkoski Foster, 2008; Marsiglio, 1991; Sayer et al., 2004; Sevilla Borra, 2015). Better educated parents spend more time in educational childcare activities, including reading, playing, and helping with homework (Gimenez-Nadal Molina, 2013; Hill Stafford, 1985; Kalenkoski Foster, 2008; Sayer et al., 2004), that promote the development of human capital of children (Brooks-Gunn et al., 2002).

R: We appreciate the need to discuss this issue in further detail. We extended Figure 3 to show the differences in parental investments by maternal education (years) and child gender, to address this point. Despite the raw differences in time investments by educational level, when accounting for all other factors, the increase of parental time investments to education years in our sample is quite small (0.2% increase in time investments for a 10% increase in education years). 

5. It is feasible to include in the analysis a variable indicating the presence or number of other household members (other than the father, mother or younger siblings of the children analyzed). Delgado and Canabal (2006) and Fuller, Holloway, and Liang (1996) show that members of Hispanic families allocate their time according to their strong family orientation (focusing on the family group), and parents tend to use other family members to care for their children.

R: Thank you for this comment. As noted in Tables 3 and 4, we are including in the analysis a variable measuring the total number of child caregivers in the household (self-reported), including any extended family participating in the care of the children. Since we are already accounting for the number of siblings, we cannot include total household size, as there is high multicollinearity. Our analysis shows that time investments increase about 3%, on average, with every additional caregiver in the household.

6. Why are individuals' wages and non-labor income not taken into account in the explanatory variables? Research has shown that higher wages lead to a better position at home as it increases the bargaining power within the couple (Bourguignon, Browning, Chiappori, 2009; Chiappori, 1988, 1992; Lundberg Pollak, 1993), and this higher bargaining power may be used to take responsibility for more rewarding childcare activities, such as playing with or reading to children. Non-labour income may also affect the time parents devote to childcare. Kalenkoski, Ribar, and Stratton (2005) show that mothers reduce the time devoted to active childcare when household income increases.

R: We understand the need for further clarification. While JUNAEB data is quite rich, since is an administrative source related to the Ministry of Education, it does not capture information on income or wages (as a representative survey normally does). The closest information we have that characterizes the socioeconomic situation of the household is the educational degree and occupational status. We have included a discussion of this point in the limitations.

7. It would be feasible to include a dummy variable of whether the children analyzed live in an urban or rural area. The female labor participation rate is lower in rural areas than in urban areas. In addition, there are other control variables, such as indigenous population and family structure, which can vary considerably from one area to another.

R: Thank you for this comment. We were already accounting for urban versus rural areas but was not reported in this version of the manuscript. We adjusted Tables to include these results. 

8. Regarding mothers' work, it is feasible to know whether they are self-employed or salaried. The literature shows a positive relationship between self-employment and time spent on childcare (Conelly 1992; Edwards and Field-Hendrey 1996; Caputo and Dolinsky 1998; Boden 1999; Gimenez et al., 2013; Campaña et al., 2020).

R: We appreciate the need to further classify the nature of female labor supply. We separated salaried from self-employed work for both parents and included into the analysis of time investments.

---

## [Decision Letter · Decision Letter 1]

21 Jun 2023

Distributional effects of parental time investments on children's socioemotional skills and nutritional health

PONE-D-23-03208R1

Dear Dr. Caro,

We’re pleased to inform you that your manuscript has been judged scientifically suitable for publication and will be formally accepted for publication once it meets all outstanding technical requirements.

Kind regards,

José Alberto Molina

Academic Editor

PLOS ONE

Additional Editor Comments (optional):

Reviewers' comments:

Reviewer's Responses to Questions

**Comments to the Author**

1. If the authors have adequately addressed your comments raised in a previous round of review and you feel that this manuscript is now acceptable for publication, you may indicate that here to bypass the “Comments to the Author” section, enter your conflict of interest statement in the “Confidential to Editor” section, and submit your "Accept" recommendation.

Reviewer #1: All comments have been addressed

Reviewer #2: All comments have been addressed

2. Is the manuscript technically sound, and do the data support the conclusions?

Reviewer #1: Yes

Reviewer #2: Yes

3. Has the statistical analysis been performed appropriately and rigorously? 

Reviewer #1: Yes

Reviewer #2: Yes

4. Have the authors made all data underlying the findings in their manuscript fully available?

Reviewer #1: No

Reviewer #2: Yes

5. Is the manuscript presented in an intelligible fashion and written in standard English?

Reviewer #1: Yes

Reviewer #2: Yes

6. Review Comments to the Author

Reviewer #1: The author has addressed my minor concerns. I consider that the manuscript is ready to be published.

Reviewer #2: Distributional effects of parental time investments on children's socioemotional skills and nutritional health

PONE-D-23-03208-R1

Reviewer #2

Comments

The author has correctly answered my concerns, and he has made changes in the paper considering my suggestions.

In my opinion, I would recommend the paper for publication in the Plos One journal.

7. PLOS authors have the option to publish the peer review history of their article (what does this mean?). If published, this will include your full peer review and any attached files.

Reviewer #1: No

Reviewer #2: No

---

## [Editor Report · Acceptance letter]

26 Jun 2023

PONE-D-23-03208R1 

Distributional effects of parental time investments on children’s socioemotional skills and nutritional health 

Dear Dr. Caro:

I'm pleased to inform you that your manuscript has been deemed suitable for publication in PLOS ONE. Congratulations! Your manuscript is now with our production department. 

Kind regards, 

on behalf of

Professor José Alberto Molina 

Academic Editor

PLOS ONE